# Autoimmune Limbic Encephalitis in Patients with Hematologic Malignancies after Haploidentical Hematopoietic Stem Cell Transplantation with Post-Transplant Cyclophosphamide

**DOI:** 10.3390/cells12162049

**Published:** 2023-08-11

**Authors:** Bu Yeon Heo, Myung-Won Lee, Suyoung Choi, Yunju Jung, Thi Thuy Duong Pham, Yunseon Jang, Jung-Hyun Park, Sora Kang, Jeong Suk Koh, Deog-Yeon Jo, Jaeyul Kwon, Ik-Chan Song

**Affiliations:** 1Department of Medical Science, College of Medicine, Chungnam National University, 266 Munhwa-ro, Jung-gu, Daejeon 35015, Republic of Korea; 2Brain Korea 21 FOUR Project for Medical Science, College of Medicine, Chungnam National University, 266 Munhwa-ro, Jung-gu, Daejeon 35015, Republic of Korea; 3Department of Internal Medicine, College of Medicine, Chungnam National University, 282 Munwha-ro, Jung-gu, Daejeon 35015, Republic of Korea; 4Translational Immunology Institute, College of Medicine, Chungnam National University, 266 Munhwa-ro, Jung-gu, Daejeon 35015, Republic of Korea

**Keywords:** allogeneic hematopoietic stem cell transplantation, autoimmune limbic encephalitis, cyclophosphamide, regulatory T cells, CD25, Foxp3, IL-6, fever, acute graft-versus-host disease, cytokine release syndrome

## Abstract

Autoimmune limbic encephalitis (LE) is a rare, but devastating complication of allogeneic hematopoietic stem cell transplantation (HSCT). There is currently limited evidence describing the risk factors, laboratory features, and underlying mechanisms of this neurologic adverse event. We retrospectively reviewed available clinical, imaging, and laboratory data from adult patients with hematological malignancies who underwent haploidentical HSCT with post-transplant cyclophosphamide (PTCy) at Chungnam National University Hospital from June 2016 to May 2020. Patients who developed LE were compared to those who did not based on clinical assessment, serum inflammatory biomarkers, and reconstitution of various T cell populations. Of 35 patients, 4 developed LE. There were no differences in patient demographics, donor demographics, or treatment conditions between patients that did and did not develop LE. Overall, patients with LE had worse clinical outcomes and overall survival than those without. In addition, they tended to have higher markers of systemic inflammation in the early post-transplant period, including fever, C-reactive protein (CRP), and cytokines. Remarkably, baseline interleukin-6 levels before HSCT were found to be higher in patients who developed LE than those who did not. In addition, analysis of T cell subsets showed impaired expansion of CD25+FOXP3+ regulatory T (Treg) cells in LE compared to non-LE patients despite appropriate reconstitution of the total CD4+ T cell population. Patients that developed LE within the first 30 days of HSCT were likely to have high serum IL-6 among other inflammatory cytokines coupled with suppression of regulatory T cell differentiation. Further work is needed on the mechanisms underlying impaired Treg expansion following HSCT and potential therapies.

## 1. Introduction

Allogeneic hematopoietic stem cell transplantation (HSCT) is a valuable curative therapy for hematological malignancies. Haploidentical allogeneic HSCT with high-dose post-transplant cyclophosphamide (PTCy) represents an emerging alternative option for patients with hematologic malignancies when a HLA identical sibling or a matched unrelated donor is not available [1,2]. PTCy effectively eradicates alloreactive T cells after haploidentical stem cell infusion while hematopoietic stem cells are spared, thereby aiding in the prevention of acute and chronic graft-versus-host disease (GVHD) [3,4,5]. 

Despite of the use of PTCy, however, patients continue to suffer from non-relapse mortality (NRM). While typical manifestations of acute GVHD and other NRM are thought to involve the skin, gastrointestinal tract, and liver, more recent reports have suggested that symptoms arising from central nervous system derangements may be a significant complication of immune therapies such as HSCT. Of these, a recent case report identified autoimmune limbic encephalitis (LE) as a complication of allogeneic HSCT [6]. LE is an inflammatory disease of the medial temporal lobes mediated by autoantibodies such as antibodies to N-methyl-D-aspartate (NMDA)-type glutamate receptor (GluR) or human herpesvirus 6 (HHV-6) and has often been reported as a paraneoplastic syndrome [7,8,9]. The diagnosis of LE is often difficult due to the broad range of clinical symptoms and overlapping features with other neurologic disorders, limiting the characterization of risk factors for the development of LE following allogeneic HSCT. As such, there are little data describing early changes following transplantation that precede the subacute progression of LE.

Studies have shown, however, that the immediate post-transplantation course is often complicated by systemic inflammation associated with high levels of pro-inflammatory cytokines such as IL-6, TNF-α, and IFN-γ, termed cytokine release syndrome (CRS) [10]. Imus et al. reported that 17% of patients undergoing haploidentical HSCT had severe CRS of grades 3 to 5 [11,12]. CRS has been strongly linked to not only acute GVHD but also highly morbid neurologic symptoms collectively referred to as immune effector cell associated neurologic syndrome (ICANS) [13,14]. 

The progression of inflammation may be related to dysfunction in regulatory T (Treg) cells. Treg cells are a highly immunosuppressive population of CD4+ T cells characterized by high and stable expression of the interleukin (IL)-2 receptor α chain (CD25) and the master transcription factor Forkhead box protein P3 (FOXP3) [15]. Treg cells are known to be impaired in disorders of inflammation, including autoimmune diseases and GVHD [16]. Transfer of Treg cells has been shown to alleviate GVHD symptoms in both humans and animal models [17], while other studies have suggested that the rapid expansion of Treg cells within the first month of HSCT is critical for GVHD prevention [16]. However, the mechanisms underlying potential Treg dysfunction following HSCT remain poorly understood. 

The aim of this study was to retrospectively analyze available clinical and laboratory data from 35 adult patients who underwent treatment with haploidentical HSCT with PTCy for hematologic malignancies, of which four developed limbic encephalitis. We also examine the reconstitution kinetics of Treg cells and their subsets within the first 21 days of HSCT to elucidate potential mechanisms of early systemic inflammation that may contribute to the development of LE. Our findings demonstrate that current immunomodulatory therapies in haploidentical HSCT may be insufficient and encourage further investigation into means of ameliorating CRS and Treg dysfunction for the prevention of acute GVHD and early neurotoxicity. 

## 2. Materials and Methods

### 2.1. Patients and Diagnosis of Autoimmune Limbic Encephalitis

We retrospectively analyzed adult patients (age > 18 years) with refractory hematological malignancies who underwent human leukocyte antigen (HLA) haploidentical allogeneic hematopoietic stem cell transplant (HSCT) using PTCy in Chungnam National University Hospital (Daejeon, Republic of Korea) between June 2016 and May 2020. We excluded those who received additional transplantations and those with refractory disease. Cyclophosphamide was given on post-transplant days 3 and 4 at a dose of 50 mg/kg (PTCy). Two conditioning regimens were used (Appendix A). In the myeloablative conditioning (MAC) regimen, 3.2 mg/kg busulfan and 25 mg/m^2^ fludarabine were administered on pre-transplant days −6 to −2, along with 14.5 mg/kg cyclophosphamide on pre-transplant days −3 and −2. In the reduced intensity conditioning (RIC) regimen, 30 mg/m^2^ fludarabine was given on pre-transplant days −6 to −2, 14.5 mg/kg cyclophosphamide on pre-transplant day −6 and −5, and total body irradiation (TBI) 2 Gy on pre-transplant day −1. On day 0, all patients received granulocyte colony-stimulating factor-mobilized peripheral blood stem cells (PBSCs; target CD34+ cell count, 5 × 10^6^/kg). Tacrolimus for GVHD prophylaxis was begun on post-transplant day 5 with a target level of 5 to 15 ng/mL, along with myclophenolate mofetil (maximum dose, 3 g per day in divided doses) for post-transplant days 5 to 35. Filgrastim 5 ug/kg was administered from post-transplant day 5 until neutrophil recovery. LE was diagnosed by characteristic MRI findings, specifically bilateral T2-hyperintesity of the medial temporal lobes on FLAIR images in patients with neurologic symptoms [18]. 

### 2.2. Clinical Assessment

Acute GVHD was graded using the modified Seattle Glucksberg criteria and chronic GVHD by reference to the revised Seattle criteria [19,20]. CMV and EBV reactivation was defined as PCR detection of viral DNA in whole blood at least once over the course of the study. Hepatic veno-occlusive disease (VOD) was defined by the 2016 classification from the European Society of Blood and Marrow Transplantation (EBMT) [21]. CRS was classified according to American Society for Transplantation and Cellular Therapy (ASTCT) criteria [14].

### 2.3. Analysis of Cytokines and Treg Cells

Plasma and peripheral blood mononuclear cells (PBMC) were obtained from whole blood using lymphocyte separation medium (Corning) via density gradient centrifugation. IL-6, TNF-α, and IFN-γ were analyzed by enzyme-linked immunosorbent assay of plasma samples. For flow cytometry, PBMCs were stained with live/dead fixable stain dye (Life Technologies) to distinguish live and dead cells. After PBS washing, cells were incubated with FITC-CD3 (BD Biosciences), PerCP-Cy5.5-CD4 (BD Biosciences), BV421-CD25 (BD Biosciences), APC-CD127 (Biolegend), and PE-Cy7-CD45RA (BD Biosciences). Cells were then fixed and permeabilized with Foxp3/Transcription Factor Staining Buffer Set (eBioscience) and further stained with PE-Foxp3 (BD Biosciences). As a way to consider the heterogeneity of the Treg compartment and analyze the property of Treg subgroups, we introduced the CD45RA marker to discriminate between antigen-experienced Treg (e.g., CD45RA−) and naïve Treg (e.g., CD45RA+) cells. T cells and their subpopulations were analyzed with a FACS BD LSR Fortessa™ X-20 flow cytometer (Becton Dickinson, Franklin Lakes, NJ, USA), and data were processed with FlowJo software ver. 10 (Tree Star, Woobum, OR, USA).

### 2.4. Statistical Analysis 

Statistical analyses of patients’ data in the tables were performed using the SPSS software ver. 26.0 (IBM Corporation, Armonk, NY, USA). Continuous variables were compared using unpaired two-tailed *t*-tests, and the chi-square test was used to assess differences in the distribution of categorical variables. Overall survival was assessed using the Kaplan–Meier method. Survival rates were compared using the log-rank test. Laboratory measurement data in the figures were analyzed using the unpaired or paired two-tailed *t*-tests using GraphPad Prism v7.02 (GraphPad, La Jolla, CA, USA). A *p*-value of <0.05 was considered to reflect significance. Spearman correlation analysis was performed to determine the linear association between acute GVHD grade and frequency of CD4 or Treg subsets. The Spearman correlation coefficient^®^ was estimated, and the significance of the association was evaluated based on *p*-value.

## 3. Results

### 3.1. Patient Characteristics and Clinical Outcomes

A total of 35 total patients were retrospectively analyzed, all of whom had undergone treatment with haploidentical hematopoietic stem cell transplantation (HSCT) and post-transplant cyclophosphamide (PTCy). Two conditioning methods were used (n = 19 myeloablative, n = 16 reduced intensity) and PTCy was administered on post-transplant days 3 and 4 (Appendix A). Four patients developed Autoimmune limbic encephalitis (LE) based on the diagnostic criteria proposed by Graus et al. (2017) [18]. Specifically, four patients developed subacute progression of seizures with epileptic or slow-wave activity observed on EEG and high signal intensity in bilateral medial temporal lobes on T2-weighted fluid-attenuated inversion recovery (FLAIR) MRI images (Figure 1A). There was no patient with seizure in the non-LE group. Median time to diagnosis was 25 days following haploidentical HSCT, and patients who developed LE displayed significantly shorter survival than those that did not (median OS, 1.5 months in LE vs. 11.0 months in non-ILE patients, *p* = 0.001, Figure 1B). 

Demographic and donor characteristics of patients diagnosed with and without LE are listed in Table 1, with additional clinical findings in LE patients in Table 2. Patient age, disease status, conditioning regimen, donor age and gender, stem cell dose, and infused CD3+ cell dose did not significantly differ between groups. However, patients with LE had a significantly higher grade of cytokine release syndrome (CRS) [median (range); 2 (2–3) vs. 1 (0–1), *p* = 0.004] and hepatic veno-occlusive disease [2 (50%) vs. 3 (9.7%), *p* = 0.030] than those without. In addition, all LE patients suffered from grade 3 to 4 acute GVHD at the time of LE diagnosis and were treated with methyl–prednisolone. Details of occurrence, involved organ and grade of acute GVHD are in Appendix A and Appendix A.

Consistent with previous reports, our patients presented with a fever of unknown origin (>38.0 °C) within the first few days following PBSC transplant (Figure 2A) [22,23]. Non-LE patients had a median Tmax of 38.5 °C that spontaneously resolved by post-transplant day 5. LE patients, however, had a significantly higher median Tmax of 39.6 °C (*p* = 0.007) that tended to last up to post-transplant day 6, with fever recurrence on days 12–18. As a marker of inflammation, plasma C-reactive protein (CRP) levels were obtained before transplant (BT), on the day of PTCy (post-transplant day 3), immediately following PTCy (post-transplant day 5), and on post-transplant day 21. While CRP levels were elevated in both groups on post-transplant days 3 and 5 relative to baseline, levels were significantly higher in LE than in non-LE patients ([median (range)] 2.29 (0.1–11.8) vs. 13.52 (11.5–15.2) at day 3, *p* = 0.0001; 4.76 (0.1–16.8) vs. 14.02 (12.2–15.2) at day 5, *p* = 0.0001) (Figure 2B)

### 3.2. Pro-Inflammatory Cytokine Levels between LE and Non-LE Patients

Given clinical evidence of systemic inflammation in the absence of infection, we hypothesized that the fever was likely a cytokine-mediated phenomenon similar to cytokine release syndrome (CRS) and examined plasma levels of pro-inflammatory cytokines IL-6, TNF-α, and IFN-γ (Figure 3). Samples were obtained from 10 non-LE and 3 LE patients before transplant (BT) and post-transplant days 3, 5, and 21. IFN-γ levels were markedly elevated in LE vs. non-LE patients on post-transplant days 3 (*p* = 0.0093) and 5 (*p* = 0.0084), which normalized by day 21. Similarly, TNF-α levels were significantly higher in LE patients on days 3 (*p* = 0.0205) and 5 (*p* = 0.0310) compared to non-LE patients. Interestingly, LE patients showed significantly elevated IL-6 not only on days 3 (*p* < 0.001) and 5 (*p* < 0.001) but also at baseline (*p* < 0.001) compared to non-LE patients. Non-LE patients also had a notable increase in IL-6 levels from baseline to day 3, which normalized by day 5.

### 3.3. Flow Cytometry of Conventional and Regulatory CD4+ T Cells

Regulatory T (Treg) cells have been suggested to be critical for establishing immune tolerance in haploidentical HSCT with PTCy [24,25]. Given the CRS-like state observed in the LE patients, we investigated whether LE development was associated with improper reconstitution of donor Treg cells. Treg cells are differentiated from CD4+ T cells and classically identified by expression of transcription factor Forkhead box P3 (Foxp3), which confers suppressive activity, and CD25, which allows for rapid response to IL-2 produced by self-reactive T-cells. Using flow cytometry (Figure 4A), we isolated CD4+CD3+ cells and identified subsets based on CD25 and Foxp3 expression, with true, activated Treg cells defined as CD4+CD25hiFoxp3+ T cells. Conventional CD4+ T cells (Tconv) were defined as CD4+CD25-Foxp3- and CD4+CD25+Foxp3- T cells, with the latter considered to be the activated phenotype. The proportions of these CD4+ T cell populations were compared between LE and non-LE patients across different time points. The proportion of activated conventional CD4+ T cells (Figure 4B, top) was significantly lower in LE patients before transplantation (*p* = 0.0023) and on post-transplant day 5 (*p* = 0.0057) but normalized by day 21. The inverse was true of non-activated conventional CD4+ T cells (Figure 4B, middle), where this subset was increased in LE patients compared to non-LE patients at BT (*p* = 0.0029) and day 5 (*p* = 0.0028) but also normalized by day 21. LE and non-LE patients demonstrated no difference in CD4+ (Appendix A), CD4+Foxp3+ (Appendix A) and CD4+CD25+Foxp3+ (Figure 4B, bottom) T cells at baseline. By post-transplant day 5, however, non-LE patients displayed significantly higher levels of Treg cells (*p* = 0.0076) compared to LE patients. By day 21, the proportion of Treg cells in non-LE patients had markedly increased from baseline; this expansion was not seen in LE patients.

We then performed a correlation analysis between these CD4+ subset proportions and the severity of acute graft-versus-host disease (aGVHD) in LE and non-LE patients. The frequency of CD25-Foxp3- cells within CD4+ T cells were inversely correlated with aGVHD severity at all time points (Figure 4C, middle.), while those of CD25+Foxp3- cells were positively correlated (Figure 4C, top). Before transplantation, there was no obvious relationship between CD4+CD25+Foxp3+ populations and aGVHD grade. On days 5 and 21, however, lower proportions of Treg cells were consistently correlated with higher degrees of aGVHD severity (Figure 4C, bottom). CD4+Foxp3+ T cells were also inversely correlated with aGVHD grade on post-transplant days 5 and 21 (Appendix A). Altogether, these data suggest that the patients who developed LE may have had impaired reconstitution of donor Treg cells with a concomitantly elevated proportion of activated conventional CD4+ T cells, where a lower proportion of Treg cells may lead to failed suppression of an acute graft-versus-host response

### 3.4. Subsets of Conventional and Regulatory CD4+ T Cell

Further subdivision of regulatory T cells may be achieved based on expression levels of Foxp3 and CD45RA with distinct phenotypes in suppressive capacity [26]. We defined the following CD4+ subsets: Foxp3intCD45RA+ cells (I, naïve/resting Treg cells), Foxp3hiCD45RA- cells (II, activated/effector Treg cells), Foxp3intCD45RA- cells (III, non-suppressive Treg cells), Foxp3lowCD45RA- cells (IV), Foxp3-CD45RA- cells (V, effector Tconv), and Foxp3-CD45RA+ cells (VI, naïve Tconv cells) (Figure 5A). The frequency of these subsets within CD4+ T cells was compared in LE vs. non-LE patients (Figure 5B). LE patients displayed significant expansion of naïve Treg cells (subset I) between BT and post-transplant day 3; this expansion was reversed, however, by day 5 without a concomitant increase in effector (subset II) or non-suppressive (subset III) Treg cells. In non-LE patients, on the other hand, the frequency of naïve Treg cells increased between post-transplant day 3 and day 5, and the subsequent drop in naïve Treg cells by day 21 was accompanied by significant expansion of effector and non-suppressive Treg cells (Figure 5B). Both effector and non-suppressive Treg cells were inversely correlated with acute GVHD severity (Figure 5C), suggesting that the absence of these populations may have contributed to an enhanced pro-inflammatory response in LE patients. Interestingly, LE patients demonstrated marked elevation of effector conventional T cells (subset V) with an associated decrease in naïve conventional T cells (subset VI) by day 21. In fact, the proportion of effector Tconv in LE patients was significantly higher than that of non-LE patients by this time point. These findings further support that enhanced activation of conventional CD4+ T cells with failed reconstitution of Treg cells may be a key mechanistic feature in post-transplant LE.

## 4. Discussion

In this study, we characterized early biologic markers of systemic inflammation in patients who developed limbic encephalitis among other signs of acute GVHD following haploidentical HSCT with PTCy. Upon clinical examination, LE patients were more likely to have severe GVHD, higher CRS grade and CRP levels, and prolonged, more extreme fevers within the first three weeks of transplant. Further investigation revealed that elevated levels of pro-inflammatory cytokines and impairment of early reconstitution of regulatory T cells may be associated with the development of more severe inflammatory adverse effects such as LE and acute GVHD.

All LE patients presented with seizures within the first 30 days of the transplant. As such, it was important to characterize early changes that may have contributed to the development of symptoms, particularly given the lack of human studies investigating that time frame. Early fever and high levels of CRP and inflammatory cytokines IL-6, TNF-α, and IFN-γ, as seen in our LE patients in the absence of infection, strongly resemble cytokine release syndrome, which may result in multiorgan dysfunction involving the brain, lung, and kidney [10,27]. CRS is well described in recipients of chimeric antigen receptor (CAR) T cell infusion and, more recently, other tumor therapies that function via the activation of immune effector cells [10,28]. Based on these previous reports, supraphysiologic cytokines are likely released by donor-derived alloreactive immune effector cells as well as endogenous monocytes [29]. In support of this, levels of pro-inflammatory cytokines (Figure 3) and clinical symptoms such as fever, vasodilation, and dyspnea (Figure 2A, Appendix A) were highest on post-transplant day 3 prior to administration of PTCy, which is given to suppress alloreactive T cells. Interestingly, we did not observe significant differences in CMV sero-positivity, prior transplant, HCT-CI score, or donor–recipient gender mismatch in patients that developed CRS versus those that did not, as reported in a retrospective multicenter study, although this may be due to a small sample size [27]. CRS, however, may be independently associated with neurotoxicity. Indeed, Imus et al. demonstrated that 13 out of 25 severe CRS patients developed encephalopathy [11]. Other studies have suggested that diffusion of cytokines and transmigration of the subsequently activated immune cells into the cerebrospinal fluid (CSF) and central nervous system (CNS) produces neurotoxicity with a high degree of mortality [30,31]. This neurotoxicity may manifest as a wide range of neurologic symptoms, including delirium, encephalopathy, aphasia, lethargy, agitation, and seizures, and is referred to by several titles in the literature, such as cell-mediated neurotoxicity syndrome (ICANS) and CAR-related encephalopathy syndrome (CRES). Currently, it is difficult to determine whether LE should be included within these syndromes, but nevertheless, LE may be associated with CRS and ICANS, often referred to as neurotoxicity. In its severe forms, ICANS may be associated with life-threatening features such as seizures. In most cases, CRS precedes ICANS, and it is understood as an initiating factor for ICANS, although the concurrent manifestation of CRS and ICANS may happen [10]. Of note, while limbic encephalitis has been reported to be in association with human herpes virus (HHV)-6 after allogenic HSCT, all patients in this study tested negative on PCR for HHV-6 [32].

Among the studied cytokines, IL-6 is already being considered as a potential biomarker for acute GVHD following allogenic HSCT, where elevated IL-6 levels were significantly associated with worsening outcomes, including severe CRS, acute GVHD, and reduced overall survival [33,34]. In our study, IL-6 levels were elevated in patients who developed LE even before HSCT (Figure 3C). It has been reported that conditioning regimens lead to host tissue damage and elevated levels of inflammatory cytokines, including IL-6, which is important in the initiation phase of acute GVHD pathophysiology [35]. LE patients showed significantly elevated IL-6 not only at baseline but also on post-transplant days 3 and 5 compared to non-LE patients. In a study with humanized NSG mice using a patient-derived leukaemic cell line, depletion of phagocytic cells prior to CAR T cell transplantation was enough to abrogate IL-6 production and CRS and, furthermore, single-cell analysis of leukocytes isolated during CRS identified monocyte lineage cells to be the source of IL-6 [33]. Through its involvement in B-cell differentiation and auto-antibody production, differentiation of Th17 cells, inhibition of Treg differentiation, or differentiation of CD8+ cytotoxic T cells, IL-6 has been reported to possibly induce and enhance autoimmune neuronal tissue damage [36,37,38]. Given that IL-6 has been strongly implicated in neuroinflammation [39], IL-6 may be a promising biomarker for screening patients at greater risk of developing neurotoxic sequelae. In addition, treatment of CRS patients in blinatumomab and CAR T cell infusion trials with the IL-6 receptor antagonist tocilizumab was shown to produce rapid clinical stabilization [12,40,41,42]. Similar therapy with tocilizumab should be further studied in HSCT to prevent severe CRS and early CNS complications. 

In addition to elevated IL-6, we also observed a significant decrease in the frequency of Treg cells in LE patients by day 5 despite an initial expansion of naïve Treg cells on day 3, suggestive of suppressed Treg cell differentiation. IL-6 is known not only to inhibit TGF-β-induced T cell differentiation into regulatory T cells [43] but also to downregulate Foxp3 expression on Treg cells [44]. Non-LE patients, on the other hand, had a significant expansion of Treg cells by post-transplant day 21. Given that Treg cells are critical for suppressing inflammatory responses and have been implicated in several autoimmune diseases [45], this early lack of reconstitution of Treg cells in LE patients may have heralded the onset of acute GVHD and neurologic symptoms. Indeed, we show that reduced Treg cells frequency was associated with greater severity of acute GVHD (Figure 4C), consistent with previous reports stating that protection against GVHD depends not only on the depletion of donor alloreactive T cells but also on the rapid and robust recovery of donor Treg cells to initiate and maintain alloimmune regulation [46,47,48,49,50,51]. Further studies have shown Treg cells from patients with acute GVHD exhibited multiple dysfunctions, including Foxp3 expression instability and increased apoptosis [52]. The underlying mechanism among early induction of IL-6, Treg cell suppression or dysfunction, and early neurotoxic clinical outcomes, such as LE, should be carefully investigated in future studies.

Despite significant associations between clinical outcomes and biological markers that may guide future HSCT research and protocol, this study has some limitations. The small sample size, low frequency of LE occurrence, and variability between patient presentation for sample collection on specific days post-transplant may mask more significant associations. Among 31 non-LE patients, PBMC samples from 10 non-LE patients could be available for the analysis of cytokines and Treg cells. We also lacked the means for autoantibody testing at this institute, which may provide additional insight into the mechanisms of LE development in future HSCT studies.

The development of LE as severe Inflammatory adverse effects after haplo-SCT may be regulated by early increased levels of pro-inflammatory cytokines such as IL-6, TNF-α, and IFN-γ and impairment of early reconstitution of regulatory T cells. The central nervous system (CNS) has been reported to be one of the nonclassical GVHD target organs that is sterile [53], and furthermore, clinical manifestations of acute GVHD in the CNS consist of signs of encephalitis [54,55,56]. Following allo-HSCT in mice, the CNS was significantly infiltrated by donor T cells, and allo-HSCT recipients with GVHD manifested cell death of neurons as a direct target of alloreactive T cells [57].

Interestingly, blockade of the IL-6 signaling resulted in marked inhibition of donor T cell accumulation, inflammatory cytokine gene expression, and host microglial cell expansion in the brain [58]. Previous preclinical and clinical studies suggest that the CNS may be targeted by donor T cells-involved alloimmune reactions [55,57,59,60]. In addition, MHC class I protein was reported to be expressed in hippocampal neurons [61,62], which suggests that the limbic system could be damaged by alloimmune reactions based on HLA mismatch, leading to the development of LE. In our study, LE patients frequently show specific HLA types, that is HLA B*40:02 and HLA DRB1*08:02 (Appendix A), suggesting that patients with these HLA types might be more susceptible to the autoimmune attack of immune effector cells. The development of LE is possibly associated with an alloimmune reaction to various proteins in the limbic system through auto-antibody formation of B cells, which may not be appropriately regulated because of the decreased function of Treg. These processes could be augmented by elevated pro-inflammatory cytokines, such as IL-6, TNF-α, and IFN-γ.

In conclusion, patients that developed LE within the first 30 days of HSCT were likely to have high serum IL-6, among other inflammatory cytokines, coupled with suppression of regulatory T-cell differentiation that may have allowed for rampant and unchecked inflammation. This association, along with potential impacts of anti-IL6 therapies, should be validated in a larger and geographically and clinically diverse population in future multicenter studies.

## Figures and Tables

**Figure 1 cells-12-02049-f001:**
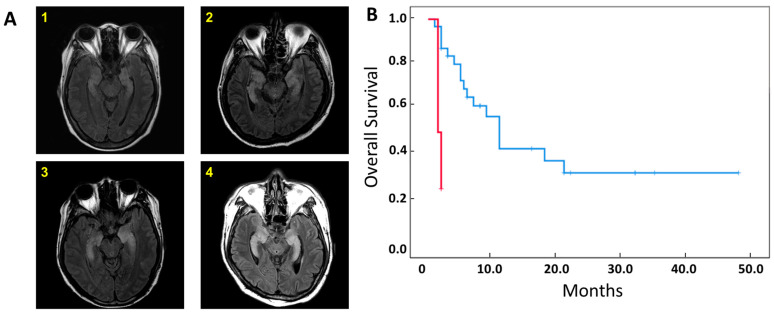
LE patients show characteristic MRI findings and reduced overall survival. (**A**) T2-weight fluid-attenuated inversion recovery (FLAIR) images in LE patients (1–4) demonstrated bilateral and symmetrical high signal intensity in both medial temporal lobes, insulae, and medial frontal lobes. (**B**) Survival curve in patients diagnosed with (red) and without (blue) LE, Statistical difference by log-rank test (N = 35, *p* = 0.001).

**Figure 2 cells-12-02049-f002:**
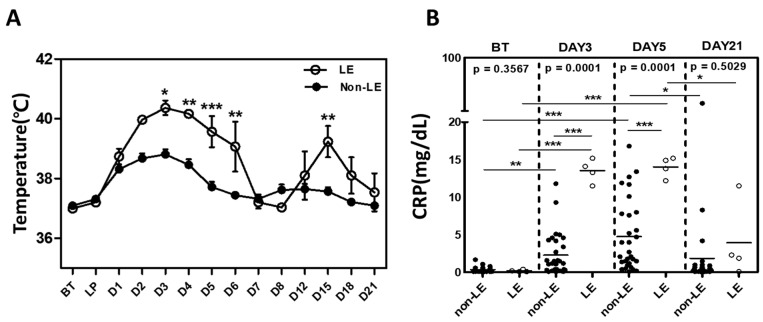
LE patients have significantly higher Tmax and plasma CRP levels. (**A**) Median Tmax in LE vs. non-LE groups based on recorded patient body temperatures up to 21 days post-transplantation. (**B**) Plasma CRP levels before transplant (BT) and days 3, 5, and 21 after haplo-identical HSCT. N = 31 non-LE patients, N = 4 LE patients. Statistical difference by two-tailed *t*-test. * *p* < 0.05, ** *p* < 0.01, *** *p* < 0.001. LP = leukapheresis.

**Figure 3 cells-12-02049-f003:**
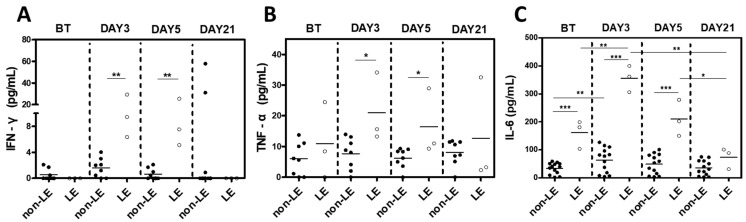
Plasma inflammatory cytokine levels are increased in LE patients. (**A**) IFN-gamma (IFN-γ), (**B**) TNF-alpha (TNF-α), and (**C**) IL-6 measured before transplant (BT) and on days 3, 5, and 21 after haplo-identical HSCT. Statistical difference by two-tailed *t*-test. * *p* < 0.05, ** *p* < 0.01, *** *p* < 0.001.

**Figure 4 cells-12-02049-f004:**
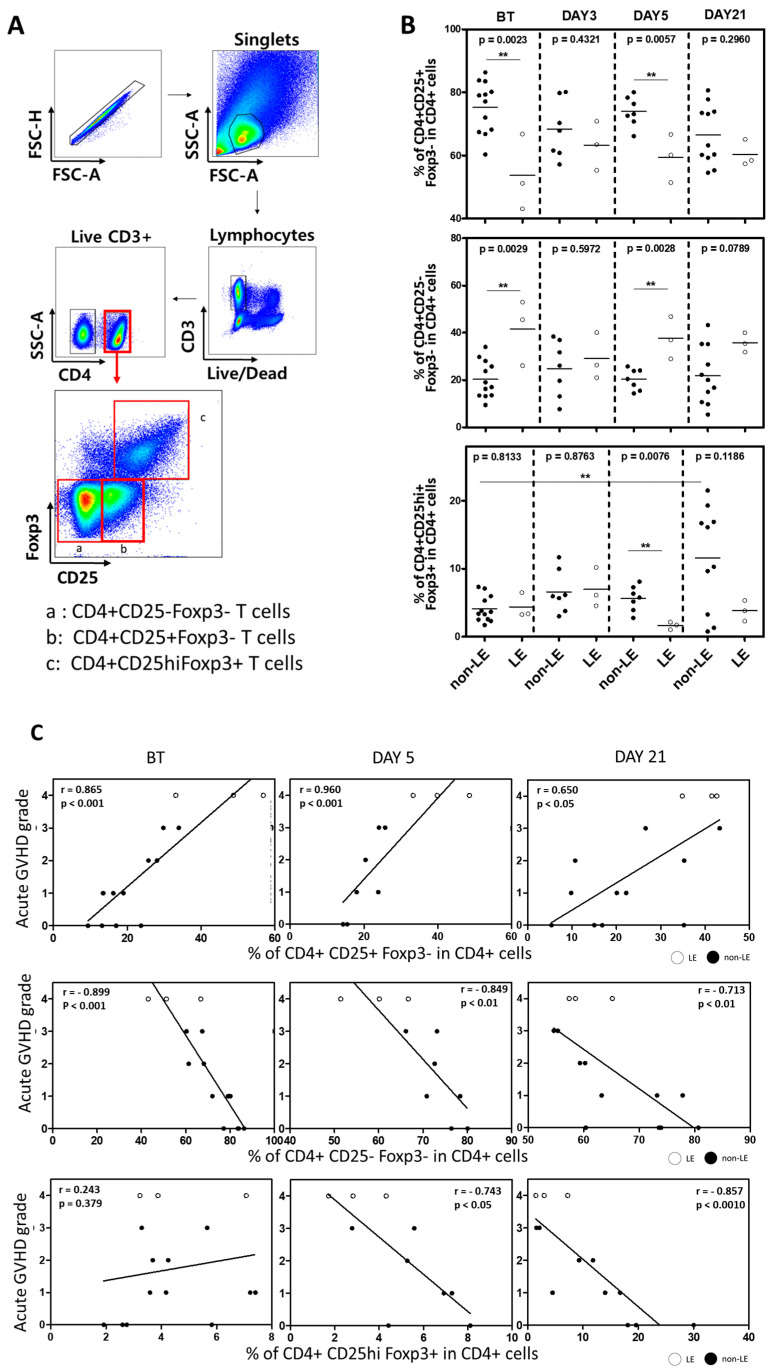
Proportion of Tregs is reduced in LE patients and is inversely related to the severity of acute graft-versus-host disease. (**A**) Flow cytometry gating strategy, with CD4+ T cells divided into three populations: (a) CD25-Foxp3-, (b) CD25+Foxp3-, (c) CD25hiFoxp3+. (**B**) Proportions of these CD4+ subsets among CD3+ T cells were compared between non-LE and LE patients before (BT) and on days 3, 5, and 21 post-HCST. Statistical difference by two-tailed *t*-test. ** *p* < 0.01. (**C**) Spearman correlation analysis between CD4+ subsets (CD25+Foxp3-, top; CD25-Foxp3-, mid; CD25hiFoxp3+, bottom) and severity of acute GVHD at three time points (BT, left; post-HCST day 5, mid; post-HCST day 21, right). Spearman correlation coefficient (r) and *p*-value were indicated.

**Figure 5 cells-12-02049-f005:**
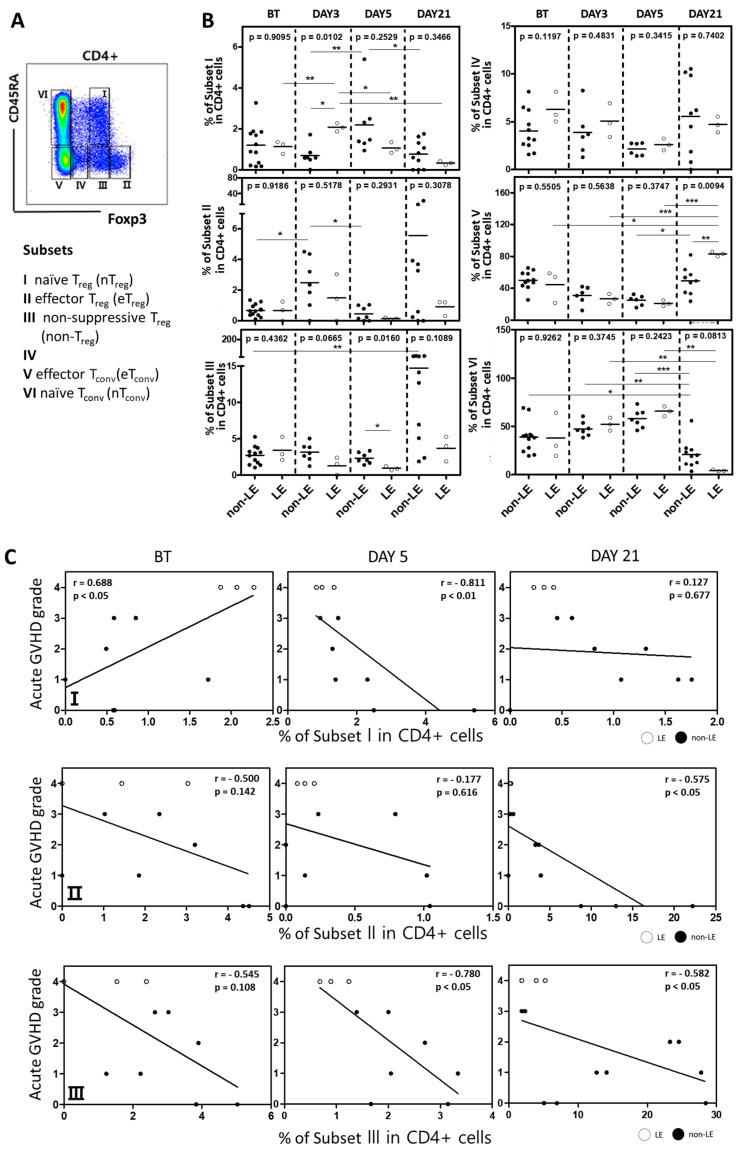
LE patients show a concomitant increase in conventional effector T cells. (**A**) Flow cytometry gating strategy with division into subsets I-VI based on CD45RA and Foxp3 as indicated. (**B**) Proportions of subsets compared between non-LE and LE patients before transplant (BT) and on days 3, 5, and 21 post-HCST. Statistical difference by two-tailed *t*-test. * *p* < 0.05, ** *p* < 0.01, *** *p* < 0.001. (**C**,**D**) Spearman correlation analysis between severity of acute GVHD and (**C**) Treg subsets (I, top; II, mid; III, bottom) (**C**) and remaining subsets (IV, top; V, mid; VI, bottom) (**D**). Spearman correlation coefficient (r) and *p*-value were indicated.

**Table 1 cells-12-02049-t001:** Patients’ characteristics (N = 35).

Limbic Encephalitis	All patients (N = 35)		*p* Value	Patients with Treg Analysis (N = 13)		*p* Value
	Patients with LE (N = 4)	Patients without LE (N = 31)		Patients with LE (N = 3)	Patients without LE (N = 10)	
**Age, median (range)**	54 (28–60)	58 (42–72)	0.429 *	56 (52–60)	60 (42–72)	0.533 *
**Gender, M:F**	3:01	14:17	0.261 ^§^	2:01	4:06	0.453 ^§^
**Disease**						
**AML**	2 (50%)	19 (61.3%)		2 (66.7%)	6 (60%)	
**ALL**	2 (50%)	5 (16.1%)		1 (33.3%)	2 (20%)	
**MDS**	0 (0.0%)	7 (22.6%)	0.22 2^§^	0 (0.0%)	2 (20%)	0.562 ^§^
**Poor Risk status ^♣^**	4 (100%)	18 (58.1%)	0.102 ^§^	3 (100%)	7 (70%)	0.290 ^§^
**Disease status**						
**1st CR**	4 (100%)	19 (61.3%)		3(100%)	5 (50%)	
**2nd CR**	0 (0.0%)	4 (12.9%)		0 (0.0%)	2 (20%)	
**MDS**	0 (0.0%)	7 (22.6%)		0 (0.0%)	2 (20%)	
**Refractory**	0 (0.0%)	1 (3.2%)	0.502 ^§^	0 (0.0%)	1 (10%)	0.466 ^§^
**Conditioning**						
**MAC**	2 (50%)	17 (57.8%)		1 (33.3%)	6 (60%)	
**RIC**	2 (50%)	14 (45.2%)	0.855 ^§^	2 (66.7%)	4 (40%)	0.453 ^§^
**HCT-CI**						
**0**	4 (100%)	15 (48.4%)		3 (100%)	6 (60%)	
**1**	0 (0.0%)	10 (32.3%)		0 (0.0%)	3 (30%)	
**2**	0 (0.0%)	2 (6.5%)		0 (0.0%)	0 (0.0%)	
**3-**	0 (0.0%)	4 (12.9%)	0.433 ^§^	0 (0.0%)	1 (10%)	0.370 ^§^
**BMI**	28 (23.4–29.4)	22.9 (18–27.4)	0.070 *	26.9 (23.4–29.1)	23.7 (18–32)	0.197 *
**Donor Gender, M;F**	4:00	24:07:00	0.288 ^§^	3:00	8:02	0.377 ^§^
**Recipient/Donor Gender mismatch**	1 (25%)	14 (45.2%)	0.443 ^§^	1 (33.3%)	7 (70.0%)	0.252 ^§^
**Donor Age**	40 (21–60)	37.5 (50–55)	0.477 *	28 (21–52)	38 (20–55)	0.559 *
**TNC (×10^8^ cells/kg)**	12.5 (9.9–14.7)	11.6 (5.9–20.4)	0.858 *	12.4 (9.9–12.6)	10.9 (5.9–20.4)	0.897 *
**CD3+cell (×10^8^ cells/kg)**	3.2 (1.7–5.1)	2.4 (1.3–4.9)	0.903 *	3.0 (1.7–3.3)	2.2 (1.0–4.9)	0.837 *
**CD34+cell (×10^6^ cells/kg)**	5.2 (2.2–9.6)	6.7 (4.8–17.7)	0.081 *	5.9 (4.5–9.6)	6.8 (4.8–17.7)	0.612 *
**CRS grade, median (range)**	2 (2–3)	1 (0–1)	0.004 ^§^	2 (2–2)	1 (0–1)	0.004 ^§^
**VOD**	2 (50%)	3 (9.7%)	0.030 ^§^	1 (33.3%)	0 (0%)	0.026 ^§^
**CMV reactivation**	2 (50%)	24 (77.4%)	0.238 ^§^	2 (66.7%)	9 (90%)	0.201 ^§^
**F/U duration, mo Median (range)**	1.7 (1.5–2.0)	5 (1–16)	0.096 *	2.0 (1.5–2.0)	5.5 (1.0–16.0)	0.089 *

^♣^ Poor risk include sAML, tAML, AML with poor risk group in NCCN guidelines, poor cytogenetics in ALL. * unpaired two-tailed *t*-test, ^§^ χ^2^ test. BMI, body mass index; CMV, cytomegalovirus; CRS, cytokine release syndrome; HCT-CI, hematopoietic stem cell transplantation-comorbidity index; MAC, myeloablative conditioning; RIC, reduced intensity conditioning; TNC, total nucleated cells; VOD, veno-occlusive disease.

**Table 2 cells-12-02049-t002:** Characteristics of 4 patients with LE.

Patients Number	Age	Sex	Disease	Initial Manifestation	Day of LE Diagnosis after HSCT	HHV-6 Status	Grade of CRS	CSF Analysis	EEG	Hepatic VOD	Grade of Acute GVHD
1	52	F	AML	Seizure	33	Negative	2	Not done	Not done	No	3
2	60	M	AML	Seizure	26	Negative	2	Not done	Epileptic wave from right anterior temporal lobe	Yes	4
3	28	M	ALL	Seizure	15	Negative	3	WBC < 5/µL	Slow-wave activity with low to medium voltage	Yes	4
4	56	M	ALL	Seizure	24	Negative	2	WBC < 5/µL	Slow-wave activity with low to medium voltage	No	3

CRS, cytokine release syndrome; CSF, cerebrospinal fluid; EEG, electroencephalogram; HHV-6, human herpesvirus 6; HSCT, hematopoietic stem cell transplantation; VOD, veno-occlusive disease; GVHD, graft-versus-host disease.

## Data Availability

All data will be made available upon request to the corresponding author.

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
