# Peer review of "Autoimmune Limbic Encephalitis in Patients with Hematologic Malignancies after Haploidentical Hematopoietic Stem Cell Transplantation with Post-Transplant Cyclophosphamide"

_cells, 2023, doi:10.3390/cells12162049_

Round 1

Reviewer 1 Report

The manuscript addresses an interesting topic of Autoimmune limbic encephalitis in patients with hematologic malignancies after haploidentical hematopoietic stem cell transplantation with post-transplant cyclophosphamide.

Although haploidentical HSCT offers a chance for cure for many patients, it can be associated with serious complications. Autoimmune limbic encephalitis is rare but severe complication following haploHSCT.  

However, some points in the manuscript need to be corrected:

The aim of the study should be presented in the Introduction section.

The order of sections in the manuscript should be changed in accordance with the recommendations of the Journal (see Instructions for authors): Introduction with the clear aim of the study; next Matherials and Metdods; Results, …

The study design is clearly presented in the Materials and Methods section. Statistical methods were correctly presented and used for analyses. The results are clearly presented, also in the form of tables and figures.   The study has its limitations like it was retrospective and single-centre, and it was conducted on a small group of patients, but these have been outlined by the authors in the Discussion.

References are mostly publications that appeared in the last 10 years.

In conclusion, the work is an important voice in the discussion on a significant complication among patients treated with haploHSCT, which is   autoimmune limbic encephalitis , although the results obtained need to be confirmed by multicentre studies on a larger group of patients.

Reviewer 2 Report

The authors present a retrospective evaluation of the rare, but devastating  complication of HSCT, namely limbic encephalitis (LE). To date, there  limited evidence describing potential risk factors, laboratory features or the underlying mechanisms of this neurologic adverse event. They retrospectively reviewed available clinicaldata, as well as imaging, and laboratory 23 adult patients with hematological malignancies after haploidentical HSCT with post-transplant cyclo- 26 phosphamide (PTCy).

This is a single center, retrospective, but well documented and highly needed analysis of PtCy complications. The involvment of CRS after PTCy and even high IL-6 levels prior to HSCT are interesting.

minor comments can be found in PDF attached
